# Simulation of Calcium Dynamics in Realistic Three-Dimensional Domains

**DOI:** 10.3390/biom12101455

**Published:** 2022-10-11

**Authors:** James Sneyd, John Rugis, Shan Su, Vinod Suresh, Amanda M. Wahl, David I. Yule

**Affiliations:** 1Department of Mathematics, University of Auckland, Auckland 1142, New Zealand; 2Department of Engineering Science, University of Auckland, Auckland 1142, New Zealand; 3Department of Pharmacology and Physiology, University of Rochester, Rochester, NY 14642, USA

**Keywords:** calcium dynamics, saliva secretion, three-dimensional simulations, finite-element methods

## Abstract

The cytosolic concentration of free calcium ions ($\left[{\mathrm{Ca}}^{2+}\right]$) is an important intracellular messenger in most cell types, and the spatial distribution of $\left[{\mathrm{Ca}}^{2+}\right]$ is often critical. In a salivary gland acinar cell, a polarised epithelial cell, whose principal function is to transport water and thus secrete saliva, $\left[{\mathrm{Ca}}^{2+}\right]$ controls the secretion of primary saliva, but increases in $\left[{\mathrm{Ca}}^{2+}\right]$ are localised to the apical regions of the cell. Hence, any quantitative explanation of how $\left[{\mathrm{Ca}}^{2+}\right]$ controls saliva secretion must take into careful account the spatial distribution of the various ${\mathrm{Ca}}^{2+}$ sources, ${\mathrm{Ca}}^{2+}$ sinks, and ${\mathrm{Ca}}^{2+}$-sensitive ion channels. Based on optical slices, we have previously constructed anatomically accurate three-dimensional models of seven salivary gland acinar cells, and thus shown that a model in which ${\mathrm{Ca}}^{2+}$ responses are confined to the apical regions of the cell is sufficient to provide a quantitative and predictive explanation of primary saliva secretion. However, reconstruction of such anatomically accurate cells is extremely time consuming and inefficient. Here, we present an alternative, mostly automated method of constructing three-dimensional cells that are approximately anatomically accurate and show that the new construction preserves the quantitative accuracy of the model.

## 1. Introduction

The concentration of cytosolic free calcium ions ($\left[{\mathrm{Ca}}^{2+}\right]$) is an important second messenger in almost all cell types, controlling a wide array of cellular functions, including contraction, movement, secretion, differentiation and gene expression [1,2,3,4]. In most cells, $\left[{\mathrm{Ca}}^{2+}\right]$ is controlled precisely in both time and space. Firstly, cells have developed mechanisms by which $\left[{\mathrm{Ca}}^{2+}\right]$ can be increased and decreased in highly restricted regions to carry out specific functions. This is accomplished principally by the extensive buffering of ${\mathrm{Ca}}^{2+}$, which decreases its effective diffusion coefficient by approximately two orders of magnitude [5] and allows for spatially localised ${\mathrm{Ca}}^{2+}$ signals [6,7,8,9]. Secondly, the signal is often carried by the frequency of oscillatory changes in $\left[{\mathrm{Ca}}^{2+}\right]$[10,11,12]. The dynamical structure of these oscillations is usually unclear, as are the precise mechanisms that control oscillation frequency. To add to the complexity, in many cells repetitive $\left[{\mathrm{Ca}}^{2+}\right]$ transients arise, not from an underlying oscillatory dynamical system but from a stochastic process [13,14,15,16].

These features make the study of $\left[{\mathrm{Ca}}^{2+}\right]$ dynamics a challenging theoretical and computational problem. In addition to the complexities inherent in oscillatory (or stochastic) processes, there are the challenges associated with spatial aspects. Often, sufficient understanding can only be obtained by performing computations in three dimensions, in domains that closely mimic those seen in real cells. Even merely constructing these domains can be a nontrivial and time-consuming problem, dependent on a large amount of data collected with considerable effort in the laboratory. It is thus important to develop ways in which such problems can be made more tractable, without losing the predictive power of the model.

Here, we consider a model of saliva secretion—which is controlled by $\left[{\mathrm{Ca}}^{2+}\right]$ oscillations—where the detailed three-dimensional structure of the cell, determined from experimental data, is a critical feature of the model. We then show how much of the intensive labour associated with such anatomically realistic computations can be bypassed by the construction of artificial three-dimensional cells with the correct overall properties.

### 1.1. Salivary Gland Acinar Cells

Saliva is secreted by the parotid, submandibular, and sublingual glands (as well as hundreds of minor glands), which consist principally of acinar and duct cells, types of polarized epithelial cells. Acinar cells in salivary glands are grouped into small clusters, like bunches of grapes, and secrete water into a common lumenal compartment to form primary saliva. This then flows through a series of tubes (much like the stem of a bunch of grapes) formed by salivary duct cells which change the saliva’s ionic composition. Primary saliva is high in ${\mathrm{Na}}^{+}$ and ${\mathrm{Cl}}^{-}$, but the secondary saliva that exits into the mouth has had much of the ${\mathrm{Na}}^{+}$ and ${\mathrm{Cl}}^{-}$ replaced by $K^{+}$ and ${\mathrm{HCO}}_{3}^{-}$ [17,18,19,20,21].

Water transport by a salivary gland acinar cell (i.e., saliva secretion; saliva is nearly all water) is the result of a two-step process: an initial ${\mathrm{Ca}}^{2+}$ response followed by the activation of ion channels and subsequent water flow via osmosis. These two processes are illustrated in the two panels of Figure 1.

The first thing to note is that each acinar cell has two different regions. The basolateral region of the cell is bounded by the basolateral membrane, which faces the interstitium. The apical region of the cell is bounded by the apical membrane and faces the lumen of the duct where the saliva is collected. The apical and basolateral membranes are bounded by tight junctions which provide a tightly controlled, although not entirely impermeable, barrier to the flow of ions and water. It is the job of the acinar cell to allow water to enter the cell across the basolateral membrane and then to allow water to flow out of the cell across the apical membrane into the lumen. In this way, water is transported from the blood to the lumen, thus producing primary saliva.

The left panel shows the principal ion channels involved in water secretion and volume control. At rest, the acinar cell accumulates ${\mathrm{Cl}}^{-}$ via the actions of the basolateral ${\mathrm{Na}}^{+}$/$K^{+}$/${\mathrm{Cl}}^{-}$ cotransporter (NKCC1) and the anion exchanger (AE4). The ${\mathrm{Na}}^{+}$ that thus enters the cell is removed by the usual ${\mathrm{Na}}^{+}$/$K^{+}$ ATPase, while, as usual, intracellular $K^{+}$ is accumulated by the NaK-ATPase. The cellular pH is controlled (at least in part) by the ${\mathrm{Na}}^{+}$/H${}^{+}$ exchanger (NHE1).

The right panel shows the principal components of the ${\mathrm{Ca}}^{2+}$ response. The binding of an agonist such as cholecystokinin (CCK) or acetylcholine (ACh) to G-protein-coupled receptors leads to the activation of phospholipase C (PLC), which cleaves phosphatidylinositol 4,5-bisphosphate into inositol trisphosphate (${\mathrm{IP}}_{3}$) and diacylglycerol. ${\mathrm{IP}}_{3}$ is an intracellular second messenger and diffuses through the cell to bind to ${\mathrm{IP}}_{3}$ receptors (IPR) on the endoplasmic reticulum. IPR are ${\mathrm{Ca}}^{2+}$ channels opened by ${\mathrm{IP}}_{3}$ and are activated quickly and inactivated more slowly by ${\mathrm{Ca}}^{2+}$. The interaction of these mechanisms leads to the periodic release and reuptake of ${\mathrm{Ca}}^{2+}$ from and to the ER, and thus the cytoplasmic $\left[{\mathrm{Ca}}^{2+}\right]$ oscillates [4].

An increase in cytoplasmic $\left[{\mathrm{Ca}}^{2+}\right]$ activates ${\mathrm{Ca}}^{2+}$-sensitive ${\mathrm{Cl}}^{-}$ channels (ClCa) on the apical membrane, leading to the flow of ${\mathrm{Cl}}^{-}$ into the lumen, whereupon water follows by osmosis. The membrane potential of the apical and basolateral membranes is maintained by ${\mathrm{Ca}}^{2+}$-activated $K^{+}$ channels (KCa) which prevent the membrane becoming so depolarised that ${\mathrm{Cl}}^{-}$ flow is hampered. Hence, agonist stimulation leads to saliva secretion.

### 1.2. The Importance of the Spatial Structure

High-resolution microscopy has shown that the IPR are located within 50 nm of the ClCa, which are located in the apical membrane. Hence, the production of ${\mathrm{IP}}_{3}$ (at the basal membrane) occurs at a site that is remote from its place of action. In addition, the concentration of the ${\mathrm{Ca}}^{2+}$ oscillation machinery in the apical region leads to localised apical ${\mathrm{Ca}}^{2+}$ oscillations that do not spread significantly to the basal region of the cell [22], so there is relatively little activation of basolateral KCa. These features raise some important questions about how the spatial structure of the cell, particularly the detailed structure of the apical membrane, affects saliva secretion. Furthermore, since all the acinar cells in a single acinus are secreting into a common lumen, there are also interesting questions about whether the arrangement of the cells within an acinus has an effect on saliva secretion.

We have previously addressed these questions by the construction of a three-dimensional model reconstructed from optical slices of a group of seven salivary gland acinar cells [23,24,25]. In these reconstructions (Figure 2) it can be seen that the apical region of the cell has a fingered shape, with each finger of the acinus being situated between two cells. We have learned a great deal from this model [26]. In particular, we know that, at least for this group of cells, apical ${\mathrm{Ca}}^{2+}$ oscillations are able to generate sufficient fluid flow over a long term and that the exact structure of the lumen appears to be of little importance. Indeed, our results suggest that, if one is concerned only with the total amount of fluid secreted by an acinus and its composition, one can dispense entirely with a three-dimensional model of the acinus and work instead with a spatially homogeneous model, with little if any loss of predictive power.

Although it might thus be tempting to immediately discard spatially homogeneous models of salivary gland acinar cells, more caution is needed. In particular, our sample size for this conclusion was n=1. Because of the time-intensive nature of the reconstruction from optical slices, it is not feasible to use this approach to check our conclusions for a wider selection of cell structures and arrangements. In addition, although this conclusion is valid if we are concerned only with the total amount and composition of primary saliva in a healthy animal, (at least for this particular structure), other questions requiring detailed three-dimensional computations may well arise in the future.

For these reasons, it is desirable to develop an automated method for the construction of three-dimensional models of groups of salivary gland acinar cells (or, indeed, any other similar cell type). It is this problem that we consider here.

## 2. Materials and Methods

### 2.1. Construction of the Artificial Cells

In previous work [27], we created artificial striated and intercalated salivary gland duct cells. We performed this by reconstructing a segment of salivary duct from a series of experimental optical slices and growing cells around this ductal skeleton, using a similar method to the one described here. This resulted in a complete ductal tube, including both intercalated and striated duct cells but with no acinus. Thus, the input to our duct model had to be computed independently and used as an input at one end of the ductal tube. Here, we add artificial acini, thus completing a complete “mini-gland” reconstruction as shown in Figure 3. This reconstruction contains 82 acinar cells (a number of which were used in the simulations reported in this work) as well as 111 duct cells (used in [27]).

Key elements in our reconstruction method are (1) spatial statistics and (2) physics-based “growing” of artificial cells from cell “seeds”. The spatial statistics are physical measurements of relevant salivary gland structural features (e.g., acinus diameter, number of cells in an acinus, etc.) that we use to guide our reconstruction process ensuring that, on average, our artificial reconstructions are representative of real-world salivary glands. In this work, we use the spatial statistics that we reported in [27], as well as physics-based “growing”, which ensures that artificially generated cells pack together tightly in a realistic organic fashion.

For this work, and in [27], we extracted a representative overall three-dimensional physical outline boundary of a mini-gland from an optical image stack containing 110 slices. However, those particular optical slices highlighted tight junctions and, as such, could not be used to accurately identify individual cell outlines. Our spatial statistics measurements were taken from an assortment of close to one hundred different optical slice images that spanned a range of linear sizes (ranging from approximately 80 to 500 μm) and resolutions (ranging from approximately 0.3 to 12 pixels/μm). The software tools that we used included Fiji [28] (for spatial measurements) and Blender [29] (for boundary outline determination and for physics-based growing). We also made extensive use of Python [30,31] as a scripting language within Blender.

#### 2.1.1. Growing Acini

The upstream beginning site of saliva production in a salivary gland consists of a group of acini sitting on top of a duct inlet. Figure 4A shows the outer boundary of a clump of six acini (in blue colour) that we extracted from optical slices. Our physics-based “growing” process was used to populate this outer boundary with tightly packed individual acini. The growth process employed the physics of elastic soft-body inflation and collision detection. Figure 4B–D show the progressive increase in size and packing associated with the growth of acini “seeds”. Note that the duct boundary (in green colour) was used as well. Growth was constrained to be both inside the acini boundary and outside the duct boundary which resulted in the tight packing shown in Panel E of Figure 4.

#### 2.1.2. Growing Acinar Cells

Since each salivary gland acinus, in turn, consists of multiple acinar cells, the next step was to grow acinar cells. Panel A in Figure 5 shows the outer boundary of one of the acini (in red colour) generated in the previous step. Again, a physics-based “growing” process was used to populate the outer boundary, this time with tightly packed individual acinar cells. Panels B through D show the progressive increase in size and packing associated with this growth. Panel E shows the final acinus with several cells removed, revealing tight cell packing including into the centre of the acinus where the acinar cell apical regions are concentrated.

#### 2.1.3. Volumetric Meshing

The finite element method employed in our simulations requires a structural partitioning of the reconstructed acinar cells into (volumetric) tetrahedra and (surface) triangles. Fortuitously, computer graphics tools, such as those we used for cell creation, typically store structural surfaces as interconnected triangle patches (known as meshes in computer graphics). We used a specialised software tool [32] to extend the two-dimensional surface mesh of each cell into a three-dimensional volumetric mesh. Figure 6 shows a cutaway view of an acinus, revealing the resultant internal tetrahedra.

#### 2.1.4. Acinus Lumen and Apical Regions

The final step in our reconstruction was to designate apical surface regions in each of the acinar cells. Note that the apical regions, taken together, form a tree-like (multiple branches, single exit) lumen structure into which primary saliva flows.

We manually created an artificial lumen tree guided by the fact that apical lumen branches coalesce towards the acinus centre, wrap each acinar cell as though with “fingers”, and generally come into contact with only two cells. A created lumen tree is shown in the top left of Figure 7. Lumen fingers can be seen wrapping around cells in the top right and bottom left of Figure 7. Finally, we used geometric distance calculations to assign as being apical the cell surface triangles that are physically close to the artificial lumen tree. The resultant apical surface triangles, with the artificial lumen tree removed, can be seen in the bottom right of Figure 7.

### 2.2. The Model

The model equations are identical to those of [27] and are thus not given in detail again here. Briefly, ${\mathrm{IP}}_{3}$ is produced only at the basolateral membrane and diffuses through the cytoplasm to bind to IPR situated so close to the ClCa on the apical membrane that all IPR ${\mathrm{Ca}}^{2+}$ fluxes are treated as boundary fluxes on the apical membrane. ${\mathrm{Ca}}^{2+}$ also diffuses through the cytoplasm but with an effective diffusion coefficient that is two orders of magnitude lower than that of ${\mathrm{IP}}_{3}$ (and the other ions). It is this low effective ${\mathrm{Ca}}^{2+}$ diffusion that allows for increases in $\left[{\mathrm{Ca}}^{2+}\right]$ that are restricted to the apical region. Previous modelling work [33,34] has made a variety of other assumptions, such as an actively propagated ${\mathrm{Ca}}^{2+}$ wave that travels from the apical to the basal region, or the presence of a mitochondrial barrier that disrupts ${\mathrm{Ca}}^{2+}$ diffusion. However, our recent experimental work [22] has shown that such intracellular ${\mathrm{Ca}}^{2+}$ waves do not exist in salivary gland cells in a living mouse, and neither is there any evidence of a mitochondrial diffusive barrier. The microstructure of the ER is taken into account only implicitly; homogenization methods are used to assume that the cytoplasm and the ER exist at each point in space [35].

The dynamics of the IPR are described by the model of [36]. ${\mathrm{Ca}}^{2+}$ oscillations that are restricted mostly to the apical region stimulate ClCa and KCa, resulting in ionic currents and thus water flow via osmosis. ${\mathrm{Na}}^{+}$, $K^{+}$, ${\mathrm{Cl}}^{-}$, ${\mathrm{HCO}}_{3}^{-}$, and H${}^{+}$ are all assumed to diffuse so fast within the cell that their concentrations can be assumed to be spatially homogeneous. The reaction–diffusion equations for ${\mathrm{Ca}}^{2+}$ and ${\mathrm{IP}}_{3}$ are solved using a finite element method, which is coupled to a system of ordinary differential equations describing the other ionic concentrations.

For our discussion here, the most important parameter is $V_{\mathrm{PLC}}$, which is the maximal rate at which ${\mathrm{IP}}_{3}$ is produced by activated PLC. Since PLC is activated by agonists binding to its receptor, $V_{\mathrm{PLC}}$ is a proxy for the level of agonist stimulation.

## 3. Results

Typical results, computed from a cell based on an anatomical reconstruction, are shown in Figure 8. The cell has a volume of 1003 $\mu m^{3}$, and the apical membrane has a fingered structure that wraps part of the way around the cell. Total fluid flow fluctuates irregularly but increases on average with stimulation level. Here, the stimulation level refers to the amount of neurotransmitter (most likely acetylcholine) secreted by neurons that are stimulating saliva secretion. This is modelled by a constant, usually treated as a bifurcation parameter, that denotes the concentration of agonists applied to the cell, which in turn controls the amount of ${\mathrm{IP}}_{3}$ that is produced at the basal membrane. At lower stimulation levels, the ${\mathrm{Ca}}^{2+}$ oscillations are restricted almost entirely to the apical region, but as the stimulation level increases, they spread a small distance towards the basal membrane. However, even at high stimulation levels there are no significant ${\mathrm{Ca}}^{2+}$ oscillations at the basal membrane. These model behaviours are all in good agreement with experimental observations [22,24].

Analogous results, computed from a simulated cell, are shown in Figure 9. The cell has a volume of 1499 $\mu m^{3}$. The qualitative properties of the fluid flow and ${\mathrm{Ca}}^{2+}$ responses are the same as those computed in a cell from an anatomical reconstruction (Figure 8). Fluid flow fluctuates but with an average that increases with stimulation, while the ${\mathrm{Ca}}^{2+}$ oscillations are restricted to the apical region.

Similar results can be computed for each of the seven anatomical cells and for each of the 14 simulated cells in our selected (simulated) acinus. However, for the remainder of the cells, we compare only the fluid flow and lumenal ionic concentrations (at t=100 s and for three values of $V_{\mathrm{PLC}}$) for each of the simulated and anatomical cells (Figure 10), as these are the main quantities of interest.

The simulated cells produce a physiologically accurate amount of primary saliva of the correct ionic composition. The ${\mathrm{Ca}}^{2+}$ responses within each simulated cell have the correct qualitative properties, being restricted to the apical region and showing oscillatory behaviour of the correct frequency for a range of intermediate stimulation levels. At higher stimulation levels, the steady-state ${\mathrm{Ca}}^{2+}$ response is a raised plateau, as seen in experiments and in the simulations using anatomical cells.

There are some quantitative differences between the responses of simulated and anatomical cells. For example, the simulated cells show a greater spread of fluid flows and lumenal ionic concentrations. However, these quantitative differences have no significant effect on the overall output of the acinus, and the results from the simulated cells are well within the physiological range.

## 4. Discussion

Saliva production is a vital physiological function and has been studied in detail by both experimentalists and theoreticians for over 50 years [17,18,37]. Most recently, experiments and models have shown how the three-dimensional structure of the acinar cells plays a critical role in the production of saliva. Earlier experiments in isolated cells had suggested that it was necessary to have an intracellular periodic ${\mathrm{Ca}}^{2+}$ wave, travelling from the apical region to the basal region, in order to activate the basolateral KCl channels and thus maintain membrane polarisation [34,38]. However, more recent data have shown, firstly, that KCl channels are also present on the apical membrane [39], as are Na-K ATPase pumps [40], and, secondly, that acinar cells in situ (i.e., in a living mouse) exhibit ${\mathrm{Ca}}^{2+}$ oscillations that are restricted to the apical region, with no intracellular apical-to-basal wave [22]. These data are consistent with observations that IPR are located within 50 nm of the ClCa channels [24].

These recent data led to the construction of a new generation of saliva secretion models, in which the apical region of each acinar cell contains most of the machinery necessary for saliva secretion [22,24,26,40]. Our simulations in three-dimensional cells that were constructed from optical slices of isolated salivary gland tissue have shown how ${\mathrm{Ca}}^{2+}$ oscillations that are restricted to the apical region of each acinar cell are sufficient to drive saliva secretion and that colocalisation of IPR and ClCa is an important feature.

Although our modelling has thus highlighted the importance of three-dimensional structures for the detailed understanding of saliva secretion, this conclusion comes with some potential problems, such as that three-dimensional domains are time consuming to construct. Even the construction of the seven anatomical cells we use here took many months, including the collection of the experimental data. There is thus a need for more automated procedures, which can be used to construct many more cells of approximately the correct physical characteristics, such that the model behaves in a similar manner in both the simulated and the artificial cells.

We presented such a method here. Even though we did not achieve full automation in our process, we were able to reconstruct a mini-gland, consisting of nearly 200 cells, in approximately 80% of the time that it took us to create the seven anatomical cells. It is primarily our cell growing technique that has enabled this upscaling by over an order of magnitude. Additional work could include automating the artificial lumen tree creation.

Although the model behaves in a qualitatively similar manner in both our anatomical and simulated cells, some questions remain. Firstly, the volumes of the simulated cells are larger on average than the volumes of the anatomical cells. All the cell volumes are based on experimental measurements, but the simulated cells are based on measurements from live mice, while the anatomical cells are based on data from an isolated gland. The causes underlying such differences are unclear; it is possible that they are due simply to random chance but equally possible that the process of isolating a gland makes the cells shrink slightly.

For the cells shown here, the different average volumes in each group of cells seem to have no significant effect. We checked this by rescaling the anatomical cells so that the mean of the anatomical cell volumes is the same as the mean of the simulated cell volumes. This resulted in minor changes to the ${\mathrm{Ca}}^{2+}$ traces but no significant changes to fluid flow or lumenal ionic composition (simulations not shown).

The most significant question that remains about the simulated cells is the source of the variability in fluid flow and lumenal ionic concentrations. When the simulated cells are recomputed, ensuring that the ratio of apical membrane area to cell volume remains constant (across all simulated cells), the variability remains, and similarly if the ratio of apical membrane area to basal membrane area remains constant (computations not shown). It is plausible that these differences in variability are simply the result of the small number of anatomical cells; the actual variability in the cell population cannot be estimated accurately from only seven cells, all taken from the same acinus. Whatever the case, the results from the simulated cells all fall well within the physiological range.

Our ultimate goal is to understand how the properties and amount of secondary saliva (i.e., the saliva that flows into the mouth, after passing through the salivary ducts) are controlled. To achieve this, we need a model of primary saliva production that is physiologically accurate. Our results here show that using simulated cells in place of anatomical cells is sufficient for this purpose.

Our method of constructing artificial cells is not restricted to only the study of salivary gland cells but is applicable to any problem requiring the use of detailed three-dimensional cellular geometries. Indeed, our method would be relatively easily extended to the construction of entire organs, as long as they had a modular structure. For example, here we constructed a group of acini, each consisting of a group of cells, all of which are organised around a given ductal skeleton. This process could be repeated to construct groups of acini, and so on, until an entire gland is constructed. At this stage, there seems little point in constructing such a large-scale model of a salivary gland, but the procedure would be easily extended to other cell types and organs, such as hepatic lobules or lung alveoli.

We have previously shown that the responses from the anatomical cells can be well approximated by a spatially homogeneous model [24], and our simulated cells have this same property. This emphasises one of the major outcomes of our previous work. In essence, if one is interested only in the total amount and composition of the primary saliva, then it is not necessary to construct a full three-dimensional model; the predictions of a vastly simpler spatially homogenous model of the apical region of the acinar cell are just as useful. However, as we pointed out before, although this holds true if one is interested only in selected macroscopic properties of the primary saliva, there is no guarantee that the same result will be true if other questions are asked. Thus, an efficient method for constructed three-dimensional cells is useful.

## Figures and Tables

**Figure 1 biomolecules-12-01455-f001:**
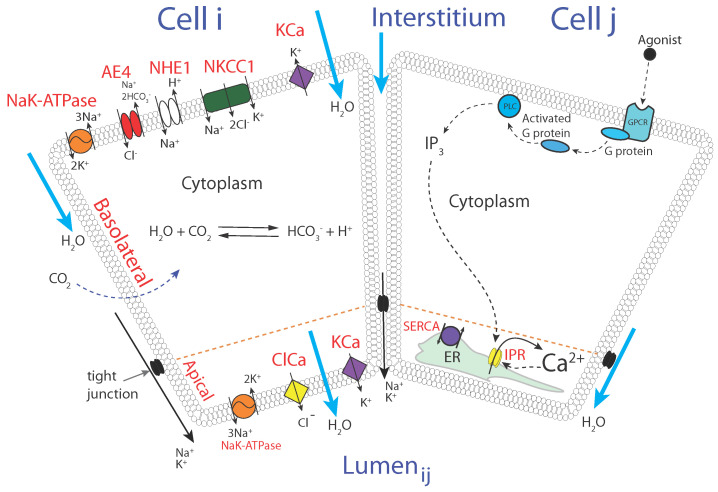
Schematic diagram of the processes underlying water transport by salivary gland acinar cells. Two identical cells are shown; cell *i* shows the relevant ion channels and transporters, while cell *j* shows the principal components of the ${\mathrm{Ca}}^{2+}$ response. Although the two sets of processes are separated here for clarity, all the processes are present in all salivary gland acinar cells. Abbreviations are PLC (phospholipase C); ${\mathrm{IP}}_{3}$ (inositol trisphosphate); IPR (${\mathrm{IP}}_{3}$ receptor); ER (endoplasmic reticulum); SERCA (sarcoplasmic/ER ${\mathrm{Ca}}^{2+}$ ATPase); KCa (${\mathrm{Ca}}^{2+}$-activated $K^{+}$ channel); ClCa (${\mathrm{Ca}}^{2+}$-activated ${\mathrm{Cl}}^{-}$ channel); NKCC1 (${\mathrm{Na}}^{+}$/$K^{+}$/${\mathrm{Cl}}^{-}$ cotransporter); NHE1 (${\mathrm{Na}}^{+}$/H${}^{+}$ exchanger); AE4 (anion exchanger). We note that these channels and exchangers are not the only ones present in the cell but are the components that are believed to be the most important for the control of water transport in this cell type. Thus, other channels and exchangers (such as, for example, RyR or Na/Ca exchangers) that are known to be critical in other cell types are not included here.

**Figure 2 biomolecules-12-01455-f002:**
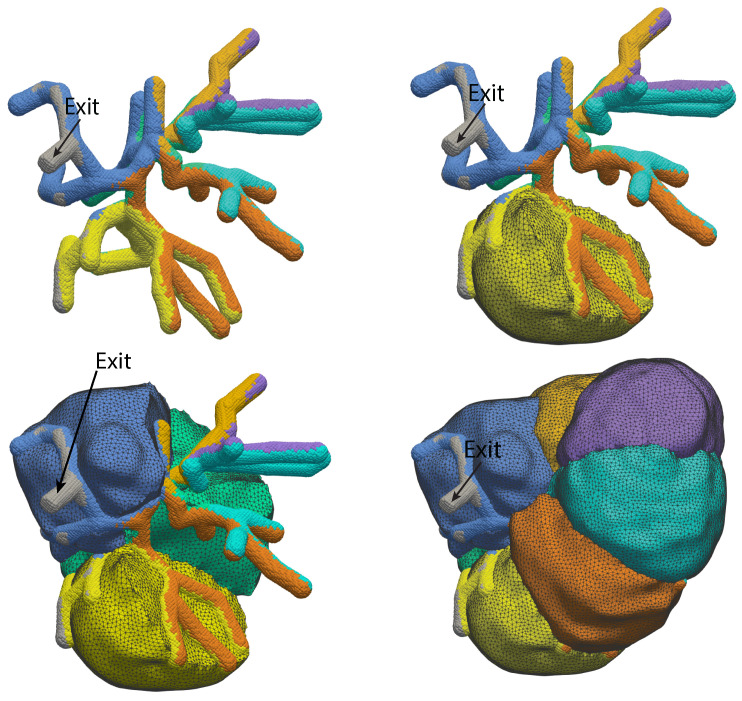
Three-dimensional reconstruction of a group of seven salivary gland acinar cells, based on data from optical slices. The panels show four different views of the same group of cells, with progressively more cells included, so that the internal structure can be more clearly seen. The top left panel shows the reconstruction of the apical lumen, with the exit to the duct shown. It is colour-coded by which cell secretes into that branch of the lumen. The top right panel shows the same lumen, but now with one acinar cell included, to demonstrate how the fingers of the lumen wrap around the cell. The bottom two panels show three and seven cells, with the same colour coding.

**Figure 3 biomolecules-12-01455-f003:**
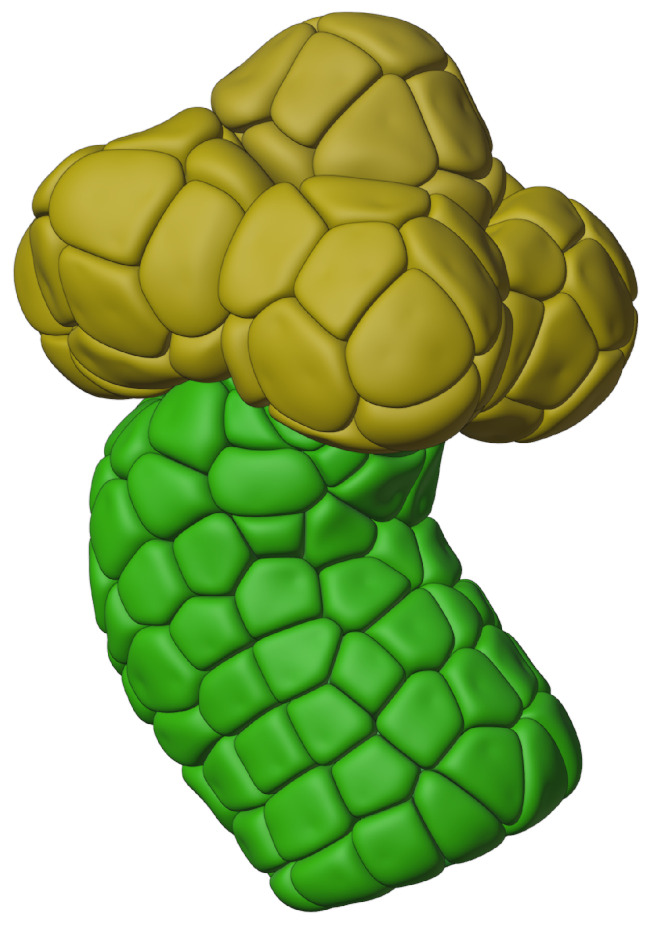
Artificial cells of a “mini-gland” subset of a salivary gland. This subset, which is replicated many times in a real salivary gland, forms the upstream beginning site of saliva production. Acinar cells are shown in yellow colour, duct cells are shown in green.

**Figure 4 biomolecules-12-01455-f004:**
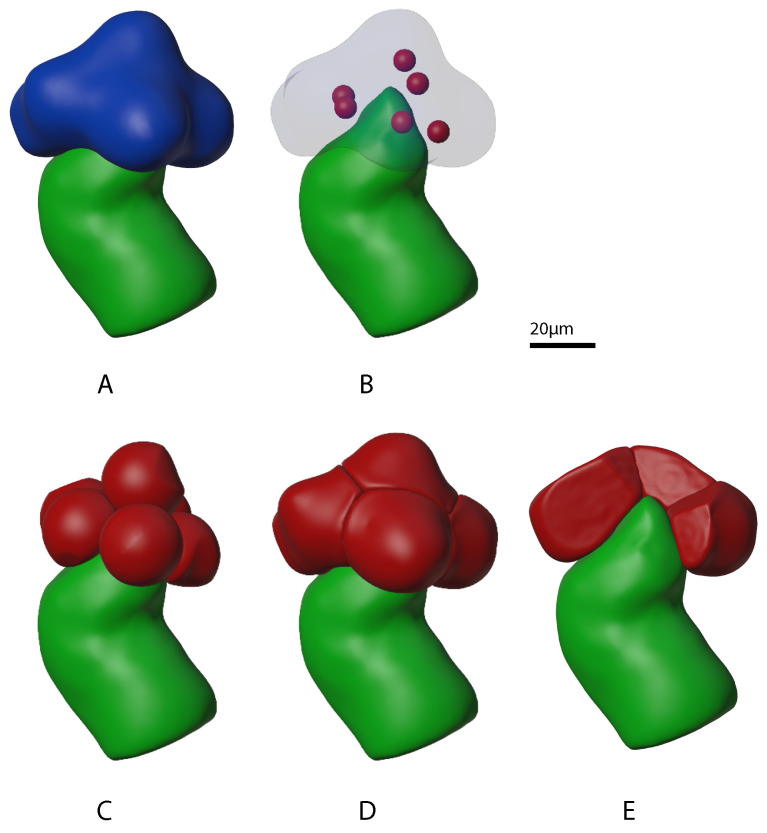
Progression of mini-gland individual acinus outer boundary “growth” stages. The mini-gland duct cell outer boundary from our prior work [27] is shown in green. (**Panel A**) shows the mini-gland acini outer boundary in blue. (**Panel B**) shows the arrangement of six individual acinus “seeds” in red. (**Panel C**) shows partially grown individual acini and (**Panel D**) shows the fully grown acini. (**Panel E**) shows the mini-gland with several of the acinus outer boundaries (red) removed. Note that the acini are packed tightly against each other as well as to the tip of the mini-gland duct outer boundary.

**Figure 5 biomolecules-12-01455-f005:**
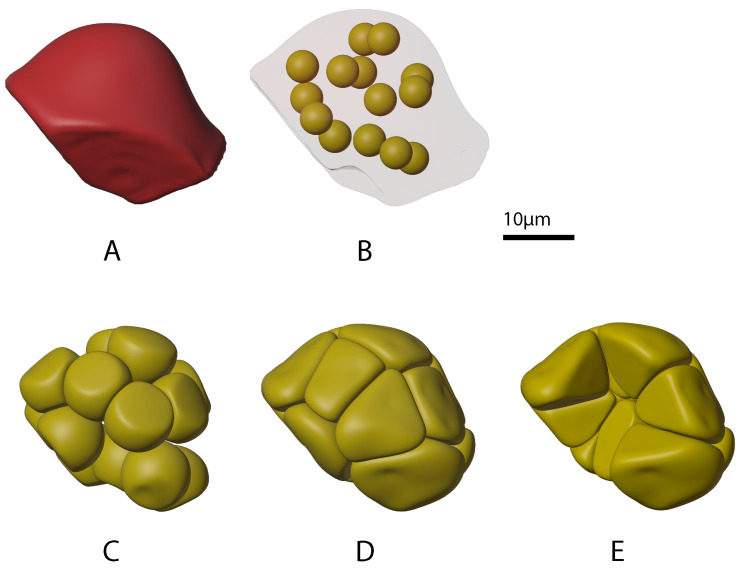
Progression of mini-gland acinus cell “growth” stages for one of the acini. (**Panel A**) shows one of the acinus outer boundaries in red colour. (**Panel B**) shows the arrangement of fourteen individual acinus cell “seeds” in yellow. (**Panel C**) shows partially grown acinus cells and (**Panel D**) shows the fully grown cells. (**Panel E**) shows the acinus with several of the cells removed. Note that the acinar cells are packed tightly together and conform to the associated acinus outer boundary.

**Figure 6 biomolecules-12-01455-f006:**
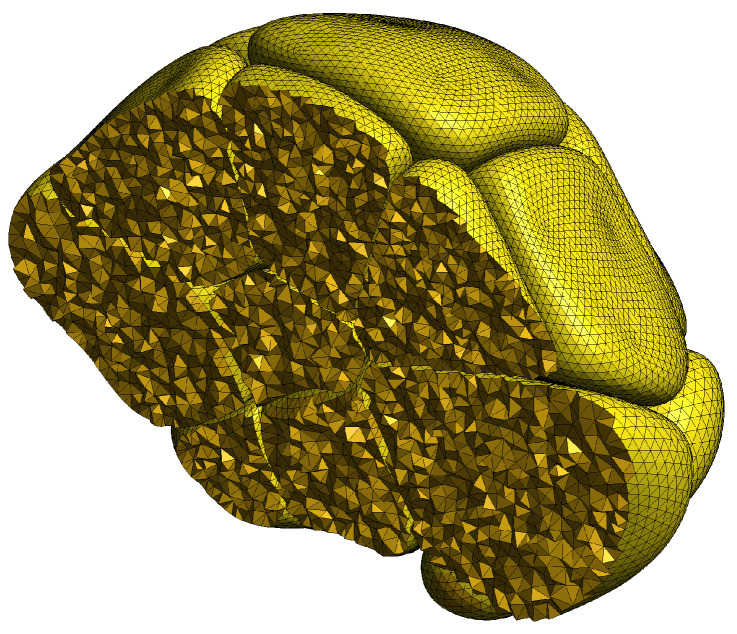
This cutaway view of a reconstructed acinus displays the volumetric tetrahedralization of each cell that is required by the three-dimensional finite element method used in our simulations. Additionally, the surface of each cell is partitioned into associated triangle patches, which are required in the simulation calculations for modelling boundary conditions.

**Figure 7 biomolecules-12-01455-f007:**
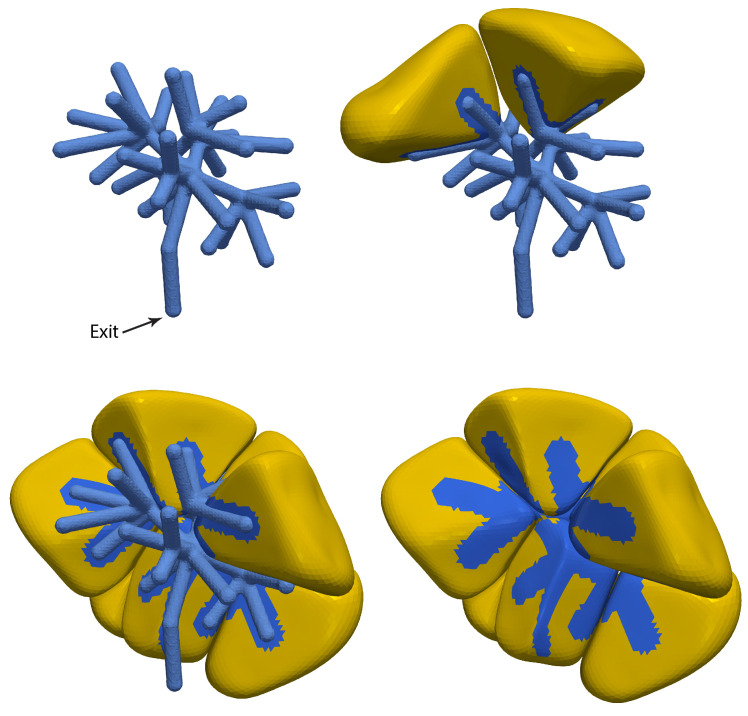
Three-dimensional reconstruction of an apical lumen “tree” associated with a 14-cell acinus. The panels show four different views of the same acinus. The (**top left panel**) shows the apical lumen reconstruction, with the exit (to the duct) labelled. The (**top right panel**) shows the same lumen, but now with two acinar cells included, to show how the fingers of the lumen wrap around each cell. Additional cells are show in the (**bottom two panels**), with the lumen tree removed in the (**bottom right panel**), fully exposing a view of the surface triangles.

**Figure 8 biomolecules-12-01455-f008:**
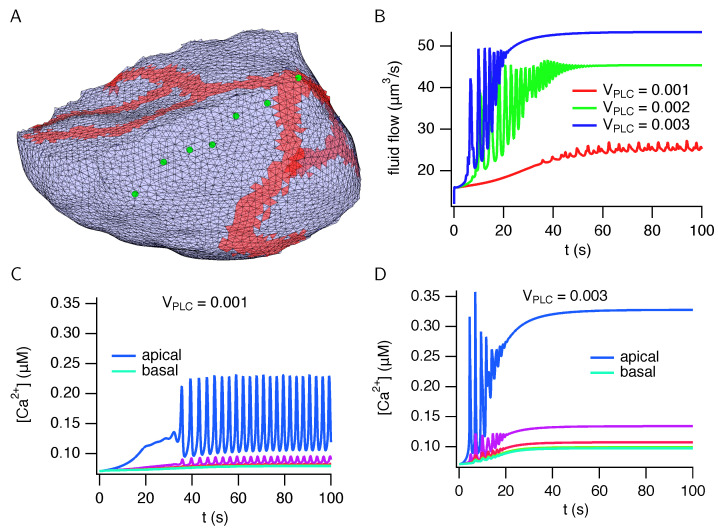
Calcium responses and fluid flow in a cell (cell 1) reconstructed from anatomical data. (**A**): the cell membrane, showing the triangles used in the finite element mesh. Red triangles are apical membrane and blue triangles are basolateral membrane. The green circles show the places for which the ${\mathrm{Ca}}^{2+}$ responses are plotted in panels (**C**,**D**). (**B**): total fluid flow plotted over time for three different stimulation levels. (**C**,**D**): ${\mathrm{Ca}}^{2+}$ responses from six different positions in the cell, denoted by the green circles in (**panel A**).

**Figure 9 biomolecules-12-01455-f009:**
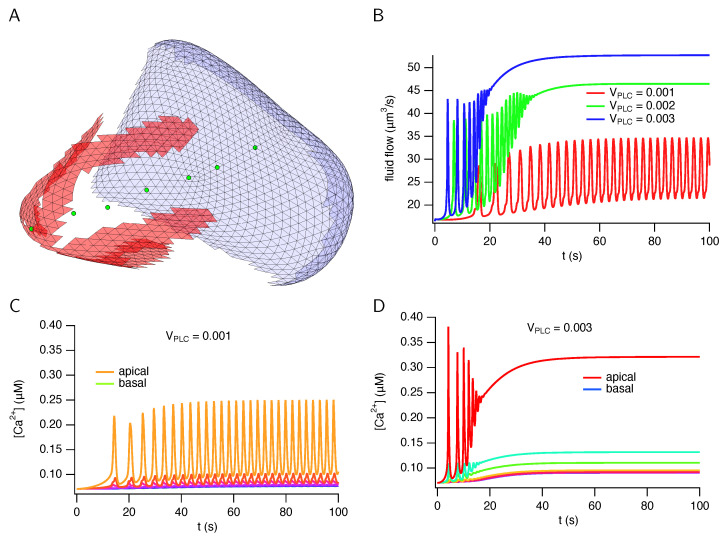
Calcium responses and fluid flow in a simulated cell. (**A**): the cell membrane, showing the triangles used in the finite element mesh. Red triangles are apical membrane and blue triangles are basal membrane. The lateral membrane (which abuts neighbouring cells) is omitted for clarity. The green circles show the places for which the ${\mathrm{Ca}}^{2+}$ responses are plotted in panels (**C**,**D**). (**B**): total fluid flow plotted over time for three different stimulation levels. (**C**,**D**): ${\mathrm{Ca}}^{2+}$ responses from seven different positions in the cell, denoted by the green circles in (**panel A**).

**Figure 10 biomolecules-12-01455-f010:**
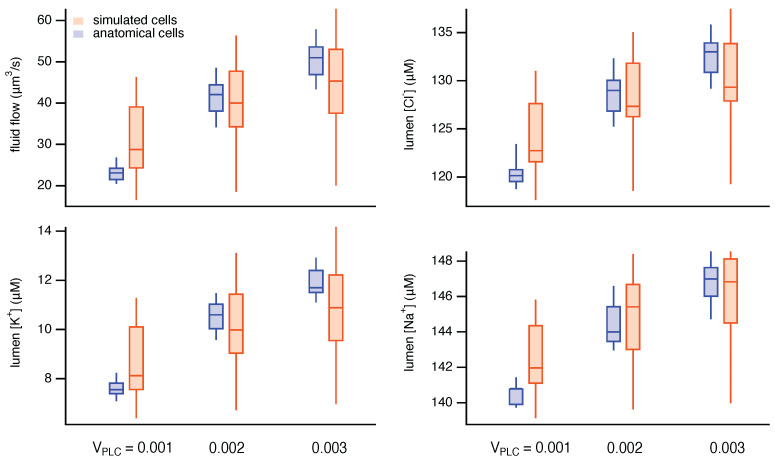
Steady-state fluid flow and lumenal ionic concentrations in simulated and anatomical cells in response to three stimulation levels. The horizontal line within each box shows the median of the relevant group, the vertical lines (the “whiskers”) extend to the maximum and minimum point in each group, while the box extends from the first quartile to the third quartile.

## Data Availability

Not applicable.

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
