# Peer review of "Simulation of Calcium Dynamics in Realistic Three-Dimensional Domains"

_biomolecules, 2022, doi:10.3390/biom12101455_

Round 1

Reviewer 1 Report

The paper og Sneyd et al., propose an automated method to generate a 3D model of salivary gland. Overall the work is well designed and presented. I have only some questions for the authors:

1. In the methods section is detailed but some informations are missing. For example at line 136 they wrote "Note that our spatial statistics measurements were taken from an assortment of different optical slices that spanned a range of sizes and resolutions". How much slices are necessary for a reconstruction? What is the range of size and resolution? Moreover, what is the software used for the reconstruction?

2. How did they measure the uM of Ca2+ showed in fig 8 and 9?

3. They claimed that this automatic model takes "less time", but how long does it thake from collection of the data to obtain the complete model?

4. Finally, the authors could explain better the future applications of the model. They said that it perfectly answer to their question, but what about other applications? the method they used to generate that model could be used for the reconstruction of other organs, glands, vessel, or other?

Thank you

Author Response

Response to Reviewer 1

  1. In the methods section is detailed but some informations are missing. For example at line 136 they wrote "Note that our spatial statistics measurements were taken from an assortment of different optical slices that spanned a range of sizes and resolutions". How much slices are necessary for a reconstruction? What is the range of size and resolution? Moreover, what is the software used for the reconstruction?

We have now addressed these questions on new pages 5 and 6, including some additional references.

  1. How did they measure the uM of Ca2+ showed in fig 8 and 9?

Figures 8 and 9 are model simulations. However, intravital measurements (i.e., measurements in a living mouse) have been made from the submandibular salivary gland, using the methods described in Takano, T. et al. Highly localized intracellular Ca2+ signals promote optimal salivary gland fluid secretion. Elife 10, e66170 (2021). Briefly, mouse were bred natively expressing the fluorescent calcium indicator GCamp6F. The submandibular gland was then exposed in an anaesthetized mouse, the brain was stimulated by electrical pulses, and the calcium responses measured by observing changes in the fluorescence. It is a new and difficult technique, but one that has demonstrated that calcium responses in live animals can be significantly different from those measured in isolated cells, or even slice preparations.

  1. They claimed that this automatic model takes "less time", but how long does it thake from collection of the data to obtain the complete model?

We add a sentence describing the time difference more precisely (new page 12). We express this time difference in relative terms only, as the actual time taken depends on the number of people assigned to the task, and how good they are.

  1. Finally, the authors could explain better the future applications of the model. They said that it perfectly answer to their question, but what about other applications? the method they used to generate that model could be used for the reconstruction of other organs, glands, vessel, or other?

We extend the discussion (new pages 12 and 13) to address these questions raised by the reviewer.

Reviewer 2 Report

This is the latest in a series of articles by this group, all devoted to the computational modeling of saliva production by a population of salivary acinar cells, requiring modeling of calcium oscillations and total electrolyte homeostasis within the cells. The particular contribution of this manuscript is an "artificial growth" algorithm to reproduce 3D structure of acinar cell clusters, which matches very closely the spatial characteristics of real acinar cells. Although some manual effort is required to reproduce the acinar lumen structure, the “artificial cell growth” algorithm is almost automated, and vastly more efficient than reconstructing 3D geometry of real cells using imaging techniques.   

Although the Authors have already applied their “artificial growth” algorithm to the reconstruction of the salivary duct cells, this is the first work where this algorithm is used to reconstruct acinar cell clusters in the salivary gland. This seems to be a logical completion of the series of previous articles by the Authors on this subject.

The Article is easy to well written, easy to follow, and concise. The presented “artificial growth” method is a powerful spatial modeling tool, deserving more exposition afforded by the present paper.

I do have several concerns below, mostly regarding the simplifying assumptions used in the modeling of electrolyte dynamics within individual acinar cells, as well as some conclusion on the role of spatial geometry. Since the paper is currently quite concise, I think there is plenty of space to explain these points clearly in the text.

Main Comments.

1)      Lines 104-109: “Indeed, our results suggest that one can dispense entirely with a three-dimensional model of the acinus and work instead with a spatially homogeneous model, with little if any loss of predictive power.  Although it might thus be tempting immediately to discard spatially homogeneous models of salivary gland acinar cells, more caution is needed.”

These two sentences seem mutually contradictory: is there a typo?

2)      Related to the statement in the 1st sentence above, on lines 222-224 the Authors mention:

“…the responses from the simulated cells can be well approximated by a spatially homogeneous model (data not shown)”.

This raises the question of the very value of exact 3D reconstruction, given that a spatially homogeneous model suffices. Perhaps the Authors mean that the surface distribution of ionic channels and other homeostasis elements is important, while the distribution of ions within the cells is not as important?  On the other hand, perhaps the validity of a homogeneous simplifications is one of results of 3D modeling; this is still fine, but should be stated somewhat more clearly, early on.

3)      Lines 193-194: “No other spatial structures (such as mitochondria, for example) are included in the cytoplasm so there are no physical barriers to the diffusion of Ca2+ and IP3.”

The assumption of unrestricted diffusion should be justified and clarified, in a couple sentences.  Is this a statement about the nature of acinar cells in particular? Cytoplasm of many cells is tightly packed with organelles. In this work, the localization of calcium by the ER and the plasma membranes is responsible for localizing oscillations to the lumen side, if I understood this correctly. Why isn’t it important to reconstruct ER (perhaps using similar "artificial growth" methods) for the 3D modeling of cell electrolyte homeostasis?

4)      Figure 1: it is striking that the Na-Ca exchanger is not present in the model, given that this is one of the primary mechanisms for calcium homeostasis in many cells, with much higher throughput compared to ATP-ase pumps (albeit with low affinity). Please explain why neglecting the Na-Ca exchanger is justified. Perhaps they are not expresses at sufficient levels in these cells? In general, I would suggest devoting a paragraph to explain why some known homeostatic processes are not included in Figure 1.

Minor Comments

1)      Lines 120-124: duct cells are only mentioned in passing. I would recommend briefly describing and summarizing the main results of the previous work by the Authors on duct cells [27], in a couple sentences.

2)      Line 202: “stimulation level”, “lower stimulation”: please describe the nature / kind of stimulation explicitly, in the context of experiment and in the model.

Author Response

Response to Reviewer 2

Main Comments.

1)      Lines 104-109: “Indeed, our results suggest that one can dispense entirely with a three-dimensional model of the acinus and work instead with a spatially homogeneous model, with little if any loss of predictive power.  Although it might thus be tempting immediately to discard spatially homogeneous models of salivary gland acinar cells, more caution is needed.”

These two sentences seem mutually contradictory: is there a typo?

We were indeed unclear. We have rephrased those sentences for greater clarity (new pages 3 and 5).

2)      Related to the statement in the 1st sentence above, on lines 222-224 the Authors mention:

“…the responses from the simulated cells can be well approximated by a spatially homogeneous model (data not shown)”.

This raises the question of the very value of exact 3D reconstruction, given that a spatially homogeneous model suffices. Perhaps the Authors mean that the surface distribution of ionic channels and other homeostasis elements is important, while the distribution of ions within the cells is not as important?  On the other hand, perhaps the validity of a homogeneous simplifications is one of results of 3D modeling; this is still fine, but should be stated somewhat more clearly, early on.

Our main point is that the validity of a homogeneous simplification is indeed one of the results of our 3D modelling, in both the anatomical and simulated cells. We have rewritten and expanded that section, and moved it to the Discussion, in an attempt to achieve greater clarity (new page 13). In fact, we now make this point twice; once, early on, on new page 3, and then again on new page 13, at the end of the Discussion.

3)      Lines 193-194: “No other spatial structures (such as mitochondria, for example) are included in the cytoplasm so there are no physical barriers to the diffusion of Ca2+ and IP3.”

The assumption of unrestricted diffusion should be justified and clarified, in a couple sentences.  Is this a statement about the nature of acinar cells in particular? Cytoplasm of many cells is tightly packed with organelles. In this work, the localization of calcium by the ER and the plasma membranes is responsible for localizing oscillations to the lumen side, if I understood this correctly. Why isn’t it important to reconstruct ER (perhaps using similar "artificial growth" methods) for the 3D modeling of cell electrolyte homeostasis?

The referee is raising a number of highly important points here.

  1. How exactly should the tortuosity of the cell cytoplasm be taken into account when modelling calcium diffusion? The short answer is that nobody knows for sure, although we do know it is certainly going to decrease the rate of diffusion.However, it is safe to say that all diffusing species (IP3, Na+, K+, Cl- and Ca2+) will be approximately equally affected by cytoplasmic tortuosity. Since Na+, K+, Cl- diffuse so quickly as to be essentially homogenous (at least on the spatial scale of a salivary gland acinar cell; we’re not talking about axons of a neuron here) this implies that tortuosity alone does not slow diffusion significantly.

The comment in the paper about “no physical barriers” was in specific reference to earlier models (Sneyd et al, Biophys. J., 85, 1392, 2003) that assumed the existence of a mitochondrial barrier to diffusion of Ca2+ from the apical to basal regions. Our current thinking is that such a mitochondrial barrier does not exist; there is certainly no evidence for it in intravital measurements.

  1. So what is it that allows Ca2+ to be high in the apical region and low in the basal region? The answer is Ca2+ buffering, which decreases the effective diffusion coefficient of Ca2+ by a factor of over 100 or more. IP3, Na+, K+, Cl- are not buffered to any significant extent and so their effective diffusion coefficient is much larger than that of Ca2+. In our model, Ca2+ buffering is taken into account by the use of a low effective Ca2+  diffusion coefficient.
  2. How does the structure of the ER affect calcium dynamics? A detailed study by Means et al (Biophys. J., 91, 537-557, 2006) showed no significant effect on dynamics of the ER geometry. Instead, the usual assumption is that the cytoplasm and ER can be modelled using homogenisation theory (Keener and Sneyd, Mathematical Physiology, Springer, 2007), and this is the approach we follow here. Of course, this assumption is not true for all cell types; cardiac cells, for example, rely on diadic clefts for their proper function. Furthermore, if the model needed to include other microdomains such as Mitochondrial Associated Membranes (MAMs) or ER/PM junctions mediating Orai/STIM interactions, then a more complicated spatial model would be required. However, that is not the case here.

To try and clarify these issues we have added some sentences on new page 8.

4)      Figure 1: it is striking that the Na-Ca exchanger is not present in the model, given that this is one of the primary mechanisms for calcium homeostasis in many cells, with much higher throughput compared to ATP-ase pumps (albeit with low affinity). Please explain why neglecting the Na-Ca exchanger is justified. Perhaps they are not expresses at sufficient levels in these cells? In general, I would suggest devoting a paragraph to explain why some known homeostatic processes are not included in Figure 1.

There is no evidence that Na/Ca exchange plays a significant role in salivary gland acinar cells. We haven’t added a specific reference for this fact in the new version (it’s not really possible to list and reference all the things that this cell type doesn’t have or use), but we can do so if the editor and reviewer consider it to be important.

Instead, we added an additional sentence to the caption of Figure 1 mentioning that many channels and exchangers (such as SOCC, RyR, Na/Ca exchanger, that don’t play a direct role in fluid secretion are omitted from this diagram. Our model (as is normally the way with models) thus presents a highly selective and simplified view of the cell.

Minor Comments

  • Lines 120-124: duct cells are only mentioned in passing. I would recommend briefly describing and summarizing the main results of the previous work by the Authors on duct cells [27], in a couple sentences.

We now do this on new page 5.

  • Line 202: “stimulation level”, “lower stimulation”: please describe the nature / kind of stimulation explicitly, in the context of experiment and in the model.

We now do this on new page 9.